# The Benefits of Caring Massage^®^ for Patients and Nurses: A Delphi Study

**DOI:** 10.3390/nursrep15020073

**Published:** 2025-02-18

**Authors:** Veronica Franchi, Jacopo Fiorini, Martina Batino, Alessandro Sili

**Affiliations:** 1Department of Biomedicine and Prevention, University of Tor Vergata, Via Montpellier, 1, 00133 Rome, Italy; vfranchi15@gmail.com (V.F.); martinabatino@gmail.com (M.B.); 2Department of Nursing Professions, University Hospital of Tor Vergata, Viale Oxford, 81, 00133 Rome, Italy; alessandro.sili@ptvonline.it

**Keywords:** caring, nurses, nursing care, patient outcome assessment

## Abstract

**Background:** The Caring Massage^®^ (CM) consists of nurse–patient physical and mental contact. It promotes empathetic presence and emotional closeness and strengthens trust and safety in the nurse–patient relationship. However, previous studies have underexplored and under-evaluated its effectiveness on different body areas. This study aimed to identify the body areas to be treated and assess CM’s influence on the nurses performing it and patients receiving it. **Method:** A Delphi study was conducted to gather expert opinions on Caring Massage^®^. A preliminary list of body areas and variables influenced by CM was developed from a literature review and submitted to panelists to identify outcomes, achieving an acceptable content validity rate. Patient and nurse variables as influenced by CM were categorized into “Bio-Physiological Outcomes” (BPOs), “Psychological Outcomes” (POs), and “Sociological Outcomes” (SOs). Two Delphi rounds were conducted between July and October 2024. **Results:** A total of 86 panelists were recruited, who identified 58 variables (7 body areas, 29 patient variables, and 22 nurse variables). Feet, legs, back, hands, shoulders, and arms were identified as key CM treatment areas. Patient outcomes identified as highly influenced by receiving CM were quality of sleep (BPO), emotional well-being (PO), and nurse–patient relationship (SO). Nurse outcomes highly influenced by performing CM were physical well-being (BPO), body respect (PO), and consideration of the patient as a person (SO). **Conclusion:** Caring Massage^®^ influences multiple aspects concerning both the patients receiving it and nurses performing it. This study addressed the heterogeneity observed in the literature, providing a foundation for future studies and encouraging further investigations.

## 1. Introduction

Although pharmaceutical and technological advances have improved treatment outcomes, patients continue to report significant dissatisfaction with their healthcare experiences [1]. Patients in vulnerable situations, such as palliative care, look for comfort and support through integrative forms of care [2].

These non-pharmacological interventions refer to health-related therapies and disciplines that are not traditionally part of medical or nursing care, and their use is increasingly incorporated into clinical practice in various forms (e.g., aromatherapy, homeopathy, acupuncture, massage, reflexology, etc.) [3]. Even in surgical settings, the use of complementary modalities is becoming more common as an adjunct treatment to reduce pain and anxiety, minimizing the use of commonly employed narcotics and anxiolytics and their associated side effects [4]. These methods are used to reduce distress, promote comfort, and foster a sense of relaxation in patients [5].

Complementary methods are also applied in specific environments, such as Intensive Care Units (ICUs), which patients often describe as “frightening” and “frustrating” [5]. These high-tech environments can contribute to stress and suffering [6]. These non-pharmacological interventions are also applied in nursing clinical practice [3], emphasizing the concept of touch, with its therapeutic and healing dimensions. It is used to reduce pain, stress, fear, depressive symptoms, and agitation and to generally enhance individual well-being [7].

### Background

Nursing embodies the concept of caring for individuals and often employs the term “Caring” to describe the professional approach of nurses in their relationship and contact with patients, implying a moral and human dimension [8]. Caring represents the act of providing care and is the essence of nursing practice, as it preserves patients’ human dignity while establishing a relationship of mutual respect, understanding, and support [9].

Based on these principles, where the mind, emotions, body, and gestures are integrated, a nursing care technique called Caring Massage^®^ has been theorized and implemented in clinical practice. Caring Massage^®^ is an intentional and mindful contact characterized by predefined touch-massage sequences performed with slow, long, rounded, and enveloping movements. The hands follow the anatomical structure of different body segments, supporting and accompanying the movements, with moderate pressure on the skin. It is a touch massage that can be offered to everyone, serving as a nursing intervention to reduce suffering, loneliness, and communication challenges [10]. Nurses, in their daily work, engage with the body and its sensations, experiences, and expressions. The act of caring for individuals in need is often mediated through touch [11].

The contact established in Caring Massage^®^ is rooted in a philosophical view of humanity and care, grounded in phenomenology and enriched by psychosocial and anatomical knowledge. Caring Massage^®^ is not intended for healing but for well-being. Its purpose is to offer the patient a moment of relief, providing them with the emotional closeness and empathy of the healthcare professional who cares for them [8].

Previous studies in the literature [2,5,12] have highlighted how touch is an integral part of nursing clinical practice, capable of fostering a sense of harmony between body and soul, as well as physical well-being and mental relaxation. Touch can mitigate suffering and pain while reducing the need for medications such as analgesics or benzodiazepines. In this context, Caring Massage^®^, implemented in studies with small sample sizes, has been shown to influence calmness, compassion, security, relief, and a sense of freedom from illness [2]. Patients feel privileged, treated with respect, and allowed to lie down and relax, temporarily setting aside their disease-related concerns and uncertainties about the future [5].

However, despite limited studies on the care-related variables and outcomes influenced by Caring Massage^®^ performed by nurses, a significant heterogeneity has been observed [1,2,6,12,13]. Researchers analyze these aspects using various qualitative and quantitative approaches, on small sample sizes, and by applying the treatment to different body areas (e.g., hands, feet, legs, face). This diversity has hindered the uniformity of evidence and the understanding of the clinical effectiveness of this treatment. Another challenge identified in the literature concerns the range of clinical care outcomes analyzed across different studies involving patients undergoing the treatment. For example, some studies assessed the effects on pain and post-traumatic stress, applying the treatment to areas such as feet, arms, and scalp [14]. Others focused on physical well-being and mental relaxation, with applications on the back, hands, and feet [2]. Some evaluated the impact on anxiety, blood pressure, and heart rate [4]. A few explored outcomes such as respect for the body and person, safety, comfort, and the nurse–patient relationship, treating the entire body [6]. Another approach grouped body areas and varied them based on treatment days [5]. Therefore, a very fragmentary analysis of this topic has emerged from the literature.

Moreover, Caring Massage^®^ also impacts the nurses who perform it. While evidence of this influence is limited, nurses report feeling closer to their patients [6], feeling more effective in alleviating patients’ suffering, and experiencing enhanced self-esteem [12]. This has led them to reassess their caregiving approach [1]. Additionally, nurses have reported a sense of calm and well-being during the treatment [12], fostering empathetic connections with patients that improve the therapeutic relationship and communication [6].

Given the limitations and heterogeneity observed in previous studies, before assessing the actual effectiveness of Caring Massage^®^, it is crucial to clarify the specific body areas to be treated and the clinical assistance outcomes influenced by this method in terms of benefits for both patients and nurses.

This clarification can help in understanding the importance of CM in nursing care, standardizing its administration and evaluating its clinical potential. Therefore, the objective of this study is to shed light on these aspects by consulting CM experts in order to support its integration into routine clinical practice. This would ensure better quality care for patients (reducing reliance on pharmacological therapies to address, for example, pain, fatigue, stress, and other conditions) and improve work environments for nurses.

## 2. Materials and Methods

### 2.1. Study Design

An observational study was conducted using the Delphi method [15] to collect the opinions of a panel of experts on the patients’ outcomes as influenced by Caring Massage^®^, aiming for a 75% consensus on each outcome [16]. This method was chosen to overcome the limits that emerged from previous studies and facilitate the measurement of the clinical efficacy of this treatment. This study was conducted and reported following the ACcurate COnsensus Reporting Document (ACCORD) guidelines (Appendix A) [17,18].

### 2.2. Sampling

A purposive sampling methodology was used to enroll experts in the field and generate robust insights and perspectives on Caring Massage^®^ [19]. To guarantee different experts’ representation, the following inclusion criteria were used: (1) certification of acquired skills and knowledge related to Caring Massage^®^ as released by the Healthcare Profession Study Center (CESPI), (2) expertise in practicing Caring Massage^®^ for at least six months, (3) educational level (Bachelor’s, Master’s, Doctorate), and (4) clinical setting and professional diversity (oncology, medicine, surgery, operating theatre; head nurses, nurses, professors). Potential participants were identified through academic publications, congress participation on this topic, and suggestions from field experts. A researcher not directly involved in the research contacted via email the potential participants, explaining the purpose, scope, and potential impact of this Delphi study. The potential participants were then invited to be part of the expert panel and their availability and informed consent to participate was requested.

### 2.3. Data Collection

A background literature review was conducted using the terms “caring touch”, “massage”, and “nurse”. From this literature review, a list of 70 possible outcomes and variables influenced by Caring Massage^®^ was created.

The investigated variables were categorized into three groups: body areas deemed most responsive to the treatment; patient clinical outcomes, focusing on anticipated benefits; and nurses’ sensations experienced during and immediately after administering the treatment to patients. Patient and nurse outcomes were organized thematically according to the World Health Organization (WHO) bio-psycho-social model, resulting in three thematic areas: Bio-Physiological Outcomes (e.g., pain, fatigue, blood pressure, pulse rate, bodily well-being, etc.), Psychological Outcomes (e.g., anxiety, stress, post-traumatic stress, mental well-being, emotional well-being, etc.), and Sociological Outcomes (e.g., nurse–patient relationship, respect for patients’ body and integrity, nurse–patient communication, organizational well-being, etc.).

A preliminary consensus was sought from two nurses experts in Caring Massage^®^ and two members of the CESPI board. The expert panel identified 66 outcomes suitable for this research (11 body area outcomes, 32 patient outcomes, 23 nurse outcomes), removing similar body area outcomes (e.g., ankles, throat, forehead, knees). A draft questionnaire was administered to the same panel as above to check comprehensibility, and the results were included in the final sample. The checked questionnaire was then delivered via e-mail and completed online by the panel of recruited nursing experts, in two rounds, between July and October 2024.

In each round, the panelists expressed their agreement on the list of potential outcomes as appropriate outcomes influenced by Caring Massage^®^, using a four-point Likert response scale (ranging from disagree = 1 to completely agree = 4). Panelists were also invited to add a comment at the end of the questionnaire and suggest additional outcomes in each round. Outcome definitions were provided to limit the potential personal interpretations. Each round was available for two weeks, and reminders were sent by email at the beginning and end of the second week. Two weeks were allowed to pass between Round 1 closing and Round 2 opening. The results from Round 1, summarized in descriptive statistics, were sent to the participants as input for Round 2. The questionnaire took approximately 5 min to complete each time, and demographic data were collected at the same time.

### 2.4. Data Analysis

The collected data were analyzed using the Statistical Package for Social Science, SPSS Ver 25^®^, and a descriptive analysis was conducted of the sociodemographic variables and the responses regarding each outcome [19]. The Likert responses of each round were converted into dichotomous variables, with scores of 1 and 2 converted to “disagree” and 3 and 4 converted to “agree”.

To evaluate the consensus, this study used response mean, standard deviation (SD), and frequencies. Percentage frequencies were calculated by dividing the frequency by the total number of participants and multiplying the result by one hundred. The data was examined also by calculating the Content Validity Ratio (CVR), the Coefficient of Variation (CV), and the Interquartile Range (IQR). CV evaluated the stability of expert responses. The interpretation of the CVR, CV, and IQR values followed the criteria set by Lawshe [20]. IQR and SD evaluated experts’ convergence, with a small value meaning high panel agreement [20]. The CVR depended on the number of panel members, and the general rule was that the numerical value of CVR based on the number of experts was higher than the acceptable values of Lawshe [20]. Assuming enrollment in each round of at least 15 experts, outcomes were considered relevant when the consensus reached ≥75% [21], the CVR was higher or equal to 0.49, and the CV was lower than 0.5 [22].

### 2.5. Ethical Considerations

This research was approved by the Institutional Review Board of the University Hospital of Rome “Tor Vergata” (protocol code 645; 30 May 2024), and it was conducted according to the principles of the Declaration of Helsinki [23]. All participants were informed about the purpose of the study and the data collection methodology. The anonymity and privacy of respondents were guaranteed (GDPR 2016/679) in each round by administering the questionnaire anonymously and presenting the results in aggregate form.

## 3. Results

### 3.1. Participants’ Characteristics

In this Delphi study, eighty-six nurse experts in Caring Massage^®^ participated in the first round, and eighty-three in the second round. Most of the experts, predominantly women (N = 76; 88.4%), were nurses (N = 64; 74.4%) with a bachelor’s degree (N = 41; 47.7%) and a specialization (N = 39; 45.3%). On average, participants had been working for 24.2 years (SD = 9.15) and had 3.5 years of experience performing Caring Massage^®^ (SD = 5.2). Table 1 summarizes the characteristics of participants of this Delphi study.

Consensus was reached in two Delphi rounds on 7 body areas, 29 patient outcomes, and 22 nurse outcomes. The patient outcomes, categorized according to the WHO bio-psycho-social model, were 7 Bio-physiological Outcomes, 14 Psychological Outcomes, and 8 Sociological Outcomes. Nurse outcomes were 1 Bio-physiological Outcome, 13 Psychological Outcomes, and 8 Sociological Outcomes.

### 3.2. Body Areas Involved

The body areas where Caring Massage^®^ was administered and evaluated by the experts through the questionnaire included 11 areas. At the end of the first round, the body areas that were eliminated were the “abdomen” (62.8%), with a CVR of 0.25 and a CV of 0.33, and the “chest” (61.6%), with a CVR of 0.23 and a CV of 0.34. In the second round, the “neck” (68.7%) had a CVR of 0.37 and a CV of 0.29, and the “scalp” (73.5%) had a CVR of 0.46 and a CV of 0.29; thus, they did not reach the required threshold to be considered.

Therefore, the body areas where Caring Massage^®^ should be performed are feet, legs, and back (97.6%; CVR = 0.95; CV = 0.13), hands (96.4%; CVR = 0.92; CV = 0.14), shoulders (95.2%; CVR = 0.9; CV = 0.17), and arms (94%; CVR = 0.87; CV = 0.16; Table 2).

### 3.3. Outcomes for Patients

The questionnaire presented 32 potential outcomes for patients as influenced by Caring Massage^®^. With the first round, three outcomes were eliminated, two of which belonged to the Bio-Physiological Outcomes category (“nausea” = 55.8%; “vomiting” = 54.7%) and one to the Psychological Outcomes category (“use of analgesics” = 72.1%), as they did not reach the consensus threshold of 75%. They had CVRs of 0.11, 0.09, and 0.44 and CVs of 0.30, 0.28, and 0.27, respectively. In the second round, the remaining outcomes were all confirmed with high percentages, and two outcomes (“emotional well-being” and “respect for the body and integrity of the person being cared for”) received full approval from the experts (100%). In the end, 29 patient-related outcomes were identified as significant for studying the effectiveness of Caring Massage^®^. Their CVRs ranged from 0.68 to 1, and their CVs from 0.10 to 0.23, indicating a high level of stability in these assessments.

We report the percentage of Bio-Physiological, Psychological, and Sociological Outcomes in Table 3 for the first and second rounds.

### 3.4. Outcomes on Nurses

The questionnaire presented 23 potential outcomes fora nurses as influenced by Caring Massage^®^. Only the outcome related to “workload perception” (55.8%), with a CVR of 0.11 and a CV of 0.31, was eliminated in the first round. All others were confirmed at the end of the second round, with approval percentages above 75%; their CVRs ranged from 0.73 to 1 and their CVs from 0.10 to 0.24, indicating a high level of stability in these assessments.

The outcomes “Respect for the body and integrity of the assisted person” and “Respect and consideration for the patient as a person” received unanimous approval from all experts (100%). Thus, 22 outcomes related to nurses providing Caring Massage^®^ were identified (Table 4).

## 4. Discussion

This study aimed to identify the body areas to be treated and the clinical outcomes influenced by Caring Massage^®^ to understand the clinical potential of this technique and its benefits for both the nurses who practice it and the patients who receive it. Fifty-eight outcomes received expert consensus, of which twenty-nine relate to the lived experience, in bio-psycho-social terms, of the patients receiving Caring Massage^®^, twenty-two concern the perceptions felt by nurses administering the treatment, and finally, seven refer to the body areas most preferable to be treated. All 58 items demonstrated high consensus, ranging from 0.66 to 1 in CVR, indicating their essential inclusion. The CV values for all items were also low, ranging from 0.10 to 0.29, indicating good agreement between the predictions or evaluations expressed by the experts [21].

Regarding the body areas, only seven were identified at the end of the Delphi process. This data can guide future studies and address the heterogeneity found in the literature. The body areas that, according to the experts, are most suitable for providing benefits to patients through Caring Massage^®^ are the feet, legs, back, hands, shoulders, and arms. They had high CVR values, ranging from 0.87 to 0.95, and low CV values, from 0.10 to 0.23, indicating a high level of stability in these assessments. The explanation for the selection of these areas and the exclusion of others can likely be found at the somatosensory level. The pleasantness of touch is correlated with the activity of slow, unmyelinated afferent fibers, known as C-tactile (CT) fibers [3,4]. CT fibers primarily innervate hairy skin and optimally respond to touch transmitted at speeds of 1–10 cm/s and at a temperature like that of human skin [24], producing a pleasant sensation, which decreases with either faster or slower touch [25,26]. Generally, most touching actions are performed with the palm of the hand coming into contact with the arm, shoulder, or back of another person [27]. While afferent CT fibers are abundantly found in hairy skin, the palm of the hand is densely innervated by fast-conducting Aβ fibers [28]. Therefore, this might suggest a preference for touch on hairy skin, as opposed to touch performed with the palm of the hand [27].

Referring to the outcomes related to patients, it is interesting to note that multiple outcomes received high consensus and generally had high CVR values, including pain (0.9), physical and mental well-being (0.92 and 0.97, respectively), stress (0.97), anxiety (0.97), feelings of loneliness (0.90), and freedom from illness (0.68). These results align with the findings in the literature. Therefore, touch massage may be effective in managing pain and anxiety in patients compared to standard care [4], in reducing levels of post-traumatic stress [12], and consequently, providing patients with a sense of bodily well-being and mental relaxation as well as relief and feelings of freedom from negative thoughts related to health and/or illness [2].

In contrast, the outcome related to “use of analgesics” did not receive consensus from the first round, presenting low CVR (0.44) and high CV (0.27) values, which suggested deletion. Therefore, the experts believed that Caring Massage^®^ is likely not fully effective in reducing patients’ need for analgesics but rather for medications to help them relax and sleep. In fact, “relaxation” and “quality of sleep” did receive consensus, with a high percentage; the CVR values were 0.95 and 0.97, with CVs of 0.14 and 0.15, respectively.

The only outcome that received a unanimous consensus from all experts was “emotional well-being”, presenting high CVR (1) and low CV (0.10) values, indicating its significant inclusion. This can likely be explained by the fact that Caring Massage^®^ fits within the concept of Caring Contact, where contact goes beyond just the physical gesture and is a true expression of compassion and empathy, with the potential to create a nurse–patient connection [6] and engaging emotions, thus nurturing the relationship of trust and improving the perceived quality of care [1].

Indeed, analyzing the consensus obtained regarding Sociological Outcomes, the experts believed that Caring Massage^®^ has the potential to most impact patients’ overall well-being and quality of life as well as the nurse–patient relationship, including communication, respect, and consideration.

These results are fully consistent with what is reported in the literature. In the study by Tengblad and colleagues [6], it emerged that patients during treatments felt more respected and considered by the nurses, which made them feel more secure. Furthermore, receiving the treatment made them feel in a state of well-being, and they sensed that they were receiving better quality nursing care [1].

Finally, the outcome related to the respect of the patient’s body and person received approval from all experts, confirming what was stated in the research of Airosa and colleagues [12], where it is mentioned that the nurse, through the contact, demonstrates presence and consideration for the patient in both body and mind.

Outcomes related to nurses for which Caring Massage^®^ has an impact showed very high percentages in the psychological and social spheres. These thematic areas had high CVR values, ranging from 0.73 to 1, and low CV values, from 0.10 to 0.24, indicating a high level of stability in these assessments. Thirteen Psychological Outcomes were considered important by most experts in both the first and second rounds. Specifically, some outcomes received the most consensus, with high content validity ratio and low coefficient of variation (CVR; CV), such as emotional well-being (0.97; 0.12), self-reflection (0.97; 0.14), mental well-being (0.97; 0.12), empathy (0.97; 0.13), tranquility (0.95; 0.15), work-related stress (0.95; 0.15), and anxiety (0.90; 0.17). This aligns with previous studies where nurses stated that during the administration of the treatment, they felt closer to the patients and more empathetic [6], experiencing the moment with a sense of calm and well-being and developing a greater awareness of themselves and others and, especially, of the emotions involved [12]. Moreover, fatigue decreased during the administration of Caring Massage^®^, as taking care of patients also meant taking care of themselves [1].

As with the outcomes related to patients, the outcome related to the “respect of the patient’s body and person” received approval from all experts, with a CVR of 1 and a CV of 0.10, indicating its significant inclusion. This aspect invites reflection on the potential of Caring Massage^®^ to also influence nurses’ approaches to engaging with patients, considering and respecting them as individuals. Indeed, all experts approved the outcome related to “respect and consideration of the patient as a person”.

Outcomes related to the “sense of professional achievement” and “nursing care satisfaction” received consensus in both rounds, confirming what is found in the literature. In the research of Airosa and colleagues [12], the perceptions of nurses are reported, with nurses saying they felt more useful because they were able to alleviate the patient’s suffering. Moreover, these feelings led them to reframe their caregiving approach, which resulted in improved job satisfaction and, therefore, in the care provided, also enhancing their sense of professional accomplishment [1].

Analyzing the outcomes with the highest consensus regarding Sociological Outcomes, it is worth reflecting on how the physical and mental proximity of the healthcare professional can be a positive aspect, not only for patients but also for the nurses themselves, as it facilitates the establishment of a trust-based relationship and improves communication. Enhancing the nurse–patient relationship nourishes self-esteem and job satisfaction for nurses [1]. This, therefore, adds value to the profession and personally enriches healthcare professionals, expanding their relational capabilities and promoting the development of a true relational style of being [29].

The results also show that Caring Massage^®^ impacts the overall well-being of nurses, which, in turn, positively affects organizational well-being. In healthcare systems, human resources play a strategic role, with a significant impact on the entire caregiving process. When healthcare professionals’ well-being declines, counterproductive behaviors can emerge at work, and consequently, the quality of care is compromised [30].

Interestingly, the outcome “workload perception” did not receive consensus in the first round, with low CVR (0.11) and high CV (0.31) values, which suggested deletion. This aspect may appear as a negative note regarding Caring Massage^®^, but in fact, it could be interpreted in a different light. Nurses who perform Caring Massage^®^ likely do not perceive this activity as an additional task to their daily caregiving duties, but rather see it as an opportunity for them to take a moment of calm, away from the often chaotic and stressful environment. This aligns with the words of a nurse from the study by Braithwaite and colleagues [1], who stated during the treatment: “It was the most relaxing part of my day”.

## 5. Limitations

The results of this study must be interpreted with consideration of several limitations. All the experts enrolled were based in Italy, and therefore, the representativeness of their expertise may be limited to this healthcare context. The reliability and validity of these results should be supported by further empirical studies in the future to effectively assess the impact of Caring Massage^®^ on clinical and caregiving outcomes for patients receiving the treatment and for nurses performing it.

## 6. Implications

Nurses and healthcare organizations must be aware of the importance of the bio-psycho-social and emotional well-being of patients to ensure, through a new caregiving approach, improved quality of care and greater support during the experience of illness and hospitalization. Additionally, given the implications of Caring Massage^®^ on nurses, this intervention could be viewed as a source of stimulation for the nursing profession, helping to foster, now more than ever, self-esteem and job satisfaction among nurses, especially considering the growing identity crisis and dissatisfaction within the profession.

## 7. Conclusions

This study identified fifty-eight outcomes as the most impacted by Caring Massage^®^. Therefore, this treatment has the potential, according to experts, to provide holistic benefits to the person, addressing all areas that make up the individual and reflecting the bio-psycho-social model of the World Health Organization (WHO). These areas were represented in this research by the Bio-Physiological Outcomes, Psychological Outcomes, and Sociological Outcomes. These three areas collectively encompass the individual outcomes related to patients receiving Caring Massage^®^ and the nurses administering it. Thus, Caring Massage^®^ not only ensures the well-being of patients but also adds “value” to the caregiving relationship and the nursing profession, enriching the healthcare professional by enhancing their empathic, communicative, and relational skills.

Moreover, according to experts, Caring Massage^®^ is more effective when performed on specific body areas, for example, the feet, legs, back, hands, shoulders, and arms, which are considered the most suitable areas to benefit from this treatment.

The results of this study aimed to clarify and address the heterogeneity found in the existing literature to provide a starting point for future studies that investigate these variables in the field. This will help to highlight the potential benefits of Caring Massage^®^ for both patients and nurses, also considering organizational and professional aspects.

## Figures and Tables

**Table 1 nursrep-15-00073-t001:** Sample characteristics.

Characteristics	N (%)	M (SD)
Age		49.5 (8.07)
Gender		
Female	76 (88.4)	
Male	10 (11.6)	
Not declared	-	
Education level		
Professional qualification	32 (37.2)	
Nurse University Diploma	13 (15.1)	
Bachelor’s degree	41 (47.7))	
Postgraduate Training		
Specialization course	60 (69.7)	
Master’s in Nursing Science	16 (18.6)	
PhD	1 (1.2)	
Work experience years		24.2 (9.15)
Caring Massage^®^ experience years		3.5 (5.2)
Current job position		
Nurse	64 (74.4)	
Nurse coordinator	9 (10.5)	
Nurse specialist	12 (13.9))	
Professor	1 (1.2)	

Note: M = Mean; SD = Standard Deviation.

**Table 2 nursrep-15-00073-t002:** Body area outcome consensus.

	Round 1 (86)					Round 2 (83)				
Body Areas	Consensus (%)	CVR	CV	M ± SD	IQR	Consensus (%)	CVR	CV	M ± SD	IQR
Feet	82 (95.3)	0.9	0.16	3.63–0.61	1	81 (97.6)	0.95	0.13	3.73–0.49	1
Legs	85 (98.8)	0.97	0.14	3.60–0.51	1	81 (97.6)	0.95	0.13	3.71–0.50	1
Abdomen	54 (62.8)	0.25	0.33	2.65–0.89	1	-				
Arms	82 (95.3)	0.9	0.16	3.64–0.57	1	78 (94)	0.87	0.16	3.59–0.58	1
Chest	53 (61.6)	0.23	0.34	2.63–0.89	1	-				1
Hands	84 (97.7)	0.95	0.16	3.60–0.57	1	80 (96.4)	0.92	0.14	3.72–0.52	1
Neck	65 (75.6)	0.5	0.33	3.00–1.00	2	57 (68.7)	0.37	0.29	3.10–0.91	2
Face	74 (86)	0.72	0.25	3.31–0.85	1	69 (83.1)	0.66	0.25	3.31–0.83	1
Scalp	70 (81.4)	0.62	0.29	3.19–0.93	1	61 (73.5)	0.46	0.29	3.10–0.91	2
Shoulders	82 (95.3)	0.9	0.19	3.53–0.69	1	79 (95.2)	0.9	0.17	3.59–0.62	1
Back	83 (96.5)	0.93	0.14	3.77- 0.54	1	81 (97.6)	0.95	0.13	3.80–0.50	1

Note: CVR = Content Validity Rate; CV = Coefficient of Variation; M = Mean; SD = Standard Deviation; IQR = Interquartile Range.

**Table 3 nursrep-15-00073-t003:** Patient outcome consensus.

Patient Outcomes		Round 1 (86)					Round 2 (83)				
		Consensus (%)	CVR	CV	M ± SD	IQR	Consensus (%)	CVR	CV	M ± SD	IQR
Bio-Physiological	Pain	74 (86)	0.72	0.22	3.27–0.73	1	79 (95.2)	0.90	0.16	3.53–0.59	1
	Fatigue	77 (89.5)	0.79	0.21	3.23–0.69	1	77 (92.8)	0.85	0.18	3.40–0.62	1
	Bodily well-being	84 (97.7)	0.95	0.15	3.53–0.54	1	80 (96.4)	0.92	0.14	3.66–0.54	1
	Blood Pressure	70 (81.4)	0.62	0.24	3.03–0.73	1	74 (89.2)	0.78	0.21	3.21–0.69	1
	Pulse rate	75 (87.2)	0.74	0.22	3.08–0.68	1	77 (92.8)	0.85	0.20	3.28–0.67	1
	Breath rate	77 (89.5)	0.79	0.21	3.13–0.68	1	76 (91.6)	0.83	0.21	3.34–0.70	1
	Quality of sleep	73 (84.9)	0.69	0.22	3.24–0.73	1	82 (98.8)	0.97	0.15	3.59–0.56	1
	Nausea	48 (55.8)	0.11	0.30	2.56–0.77	1	-				
	Vomiting	47 (54.7)	0.09	0.28	2.54–0.73	1	-				
Psychological	Anxiety	83 (96.5)	0.93	0.13	3.74–0.51	1	82 (98.8)	0.97	0.11	3.78–0.44	1
	Stress	82 (95.3)	0.90	0.15	3.68–0.55	1	82 (98.8)	0.97	0.12	3.74–0.46	1
	Post-traumatic stress	79 (91.9)	0.83	0.19	3.44–0.67	1	78 (94)	0.87	0.17	3.45–0.61	1
	Mental well-being	85 (98.8)	0.97	0.12	3.70–0.48	1	82 (98.8)	0.97	0.11	3.80–0.42	1
	Emotional well-being	85 (98.8)	0.97	0.12	3.75–0.45	1	83 (100)	1	0.10	3.79–0.40	1
	Sense of feeling lonely	78 (90.7)	0.81	0.21	3.27–0.69	1	79 (95.2)	0.90	0.18	3.38–0.62	1
	Sense of abandonment	75 (87.2)	0.74	0.23	3.25–0.76	1	77 (92.8)	0.85	0.20	3.38–0.69	1
	Sense of freedom from illness	66 (76.7)	0.53	0.25	2.97–0.76	2	70 (84.3)	0.68	0.23	3.19–0.75	1
	Tranquility	82 (95.3)	0.90	0.17	3.43–0.58	1	81 (97.6)	0.95	0.15	3.56–0.54	1
	Comfort	81 (94.2)	0.88	0.18	3.38–0.63	1	81(97.6)	0.95	0.14	3.60–0.53	1
	Relaxation	80 (93)	0.86	0.17	3.60–0.61	1	81 (97.6)	0.95	0.14	3.75–0.53	1
	Sense of security	78 (90.7)	0.81	0.20	3.24–0.64	1	77 (92.8)	0.85	0.18	3.43–0.62	1
	Compassion	64 (74.4)	0.48	0.31	3.03–0.96	2	76 (91.6)	0.83	0.21	3.36–0.70	1
	Sleep drug usage	65 (75.6)	0.51	0.27	2.93–0.80	2	72 (86.7)	0.73	0.22	3.34–0.73	1
	Analgesic usage	62 (72.1)	0.44	0.27	2.90–0.80	2	-				
Sociological	Quality of life	79 (91.9)	0.83	0.18	3.36–0.63	1	80 (96.4)	0.92	0.16	3.46–0.57	1
	General well-being	85 (98.8)	0.97	0.14	3.54–0.52	1	80 (96.4)	0.92	0.17	3.61–0.64	1
	Safety of care	75 (87.2)	0.81	0.20	3.24–0.64	1	79 (95.2)	0.90	0.16	3.48–0.59	1
	Nurse–patient relationship	84 (97.7)	0.95	0.13	3.70–0.50	1	82 (98.8)	0.97	0.11	3.78–0.44	1
	Nurse–patient communication	82 (95.3)	0.90	0.14	3.70–0.5	1	81 (97.6)	0.95	0.13	3.71–0.50	1
	Respect and consideration of the patient as a person	83 (96.5)	0.93	0.13	3.73–0.51	1	80 (96.4)	0.92	0.14	3.71–0.53	1
	Respect for the patient’s body and integrity	76 (88.4)	0.76	0.21	3.46–0.76	1	83 (100)	1	0.11	3.77–0.42	1
	Adaptation to hospitalization	76 (88.4)	0.76	0.21	3.16–0.68	1	79 (95.2)	0.90	0.17	3.43–0.58	1

Note: CVR = Content Validity Rate; CV = Coefficient of Variation; M = Mean; SD = Standard Deviation; IQR = Interquartile Range.

**Table 4 nursrep-15-00073-t004:** Nurse outcome consensus.

Nurse Outcomes		Round 1 (86)					Round 2 (83)				
		Consensus (%)	CVR	CV	M ± SD	IQR	Consensus (%)	CVR	CV	M ± SD	IQR
Bio-Physiological	Bodily well-being	77 (89.5)	0.79	0.19	3.40–0.67	1	76 (91.6)	0.83	0.18	3.51–0.65	1
Psychological	Job-related stress	77 (89.5)	0.79	0.20	3.38–0.70	1	81 (97.6)	0.95	0.15	3.57–0.54	1
	Anxiety	77 (89.5)	0.79	0.21	3.24–0.70	1	79 (95.2)	0.90	0.17	3.56–0.63	1
	Burnout	72 (83.7)	0.67	0.23	3.27–0.76	1	72 (86.7)	0.73	0.24	3.39–0.84	1
	Self-esteem	82 (95.3)	0.90	0.17	3.51–0.62	1	76 (91.6)	0.83	0.20	3.49–0.72	1
	Mental well-being	83 (96.5)	0.93	0.15	3.56–0.56	1	82 (98.8)	0.97	0.12	3.72–0.47	1
	Emotional well-being	84 (97.7)	0.95	0.14	3.62–0.53	1	82 (98.8)	0.97	0.12	3.74–0.46	1
	Self-reflection	80 (93)	0.86	0.18	3.5–0.66	1	82 (98.8)	0.97	0.14	3.59–0.51	1
	Compassion	72 (83.7)	0.67	0.26	3.22–0.85	1	72 (86.7)	0.73	0.23	3.32–0.76	1
	Empathy	81 (94.2)	0.88	0.17	3.62–0.63	1	82 (98.8)	0.97	0.13	3.67–0.49	1
	Tranquility	82 (95.3)	0.90	0.17	3.51–0.62	1	81 (97.6)	0.95	0.15	3.5–0.54	1
	Sense of professional achievement	82 (95.3)	0.90	0.17	3.59–0.62	1	78 (94)	0.87	0.16	3.65–0.59	1
	Nursing care satisfaction	82 (95.3)	0.90	0.15	3.61–0.57	1	80 (96.4)	0.92	0.16	3.65–0.59	1
	Respect for patients’ bodies and integrity	84 (97.7)	0.95	0.14	3.74–0.53	1	83 (100)	1	0.10	3.71–0.41	1
Sociological	Quality of life	79 (91.9)	0.83	0.18	3.40–0.63	1	74 (89.2)	0.78	0.21	3.40–0.71	1
	Workload perception	48 (55.8)	0.11	0.31	2.67–0.83	1	-	-	-	-	
	General well-being	82 (95.3)	0.90	0.18	3.45–0.62	1	80 (96.4)	0.92	0.17	3.53–0.61	1
	Organizational well-being	71 (82.6)	0.65	0.25	3.09–0.79	1	72 (86.7)	0.73	0.23	3.26–0.78	1
	Provide safe, effective, and efficient care	81 (94.2)	0.88	0.18	3.5–0.64	1	80 (96.4)	0.92	0.14	3.66–0.54	1
	Establish a relationship with the patient easily	85 (98.8)	0.97	0.13	3.68–0.49	1	82 (98.8)	0.97	0.12	3.75–0.45	1
	Communication with patient	84 (97.7)	0.95	0.13	3.68–0.49	1	82 (98.8)	0.97	0.12	3.75–0.45	1
	Respect and consideration of the patient as a person	84 (97.7)	0.95	0.14	3.74–0.53	1	83 (100)	1	0.10	3.78–0.41	1
	Physical and mental closeness to the patient	83 (96.5)	0.93	0.17	3.66–0.62	1	82 (98.8)	0.97	0.12	3.75–0.45	1

Note: CVR = Content Validity Rate; CV = Coefficient of Variation; M = Mean; SD = Standard Deviation; IQR = Interquartile Range.

## Data Availability

The data presented in this study are available on request from the corresponding author due to privacy or ethical restrictions.

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
