# Peer review of "The Benefits of Caring Massage® for Patients and Nurses: A Delphi Study"

_nursrep, 2025, doi:10.3390/nursrep15020073_

Round 1
Reviewer 1 Report
Comments and Suggestions for Authors
I found the topic of the research, Caring Massage (CM), to be an interesting topic. The study was well designed and research presented in a very 'understandable fashion'. I think the topic, CM, demonstrates a lost skill in nursing. In a brief literature search of the teaching of CM in the United States, I found that it is often not taught in our programs, being replaced by other contents such as increased technology, documentation requirements, etc. I agree with the author's assessment of limitations related to the experts being enrolled in Italy. Future implications may be to revisit implementing CM in nursing programs due to the documented benefit to patient and care givers. The article demonstrates the 'art of nursing' which we must claim as our own and use the outcomes to demonstrate our continued value in the healthcare team.

Thoughts: Line 201 3,5- is this 3.5 or 3-5 years? I see this commonly throughout the manuscript. I'm just used to 3.5 vs 3,5. Page 10 is Bornout the same as Burnout or is this a typo?
Author Response
Comments 1: General Impression
I found the topic of the research, Caring Massage (CM), to be an interesting topic. The study was well designed and research presented in a very 'understandable fashion'. I think the topic, CM, demonstrates a lost skill in nursing. In a brief literature search of the teaching of CM in the United States, I found that it is often not taught in our programs, being replaced by other contents such as increased technology, documentation requirements, etc. I agree with the author's assessment of limitations related to the experts being enrolled in Italy. Future implications may be to revisit implementing CM in nursing programs due to the documented benefit to patient and care givers. The article demonstrates the 'art of nursing' which we must claim as our own and use the outcomes to demonstrate our continued value in the healthcare team.
Response 1: Thank you for your valuable comments. We are happy that you have fully grasped our intentions. We believe CM can add more value to the nursing profession now than ever, not forgetting the importance of the "art of care" in a digital world.
Comments 2: Brief Summary
The research article contributes to the body of nursing literature related to Caring Massage (CM). It highlights the significance of this clinical skill to patients. It also emphasizes the ‘core skill’ that only nursing provides, making it a unique part of patient care.
Response 2: Thank you for your valuable comments.
Comments 3: Main research question
The study took an in-depth look at CM currently available in the literature.
Response 3: Thank you for your valuable comments. We have examined the latest articles on patient-nurse contact.
Comments 4: Topic
The topic of CM is relevant to the field as it demonstrates another outcome of nursing that is significant to patient care. The Delphi study was well-designed and included an adequate sample size.
Response 4: Thank you for your valuable comments.
Comments 5: Are conclusions consistent with the evidence and arguments presented and do they address the main question posed?
The study addressed the heterogeneity observed in Italian literature and provided the foundation for future studies and encouraged further investigations, especially on a global level. This is not a topic (CM) that is currently included in nursing programs in the United States. It was many years ago, but the curriculum has changed, and the topic has been replaced. The article provides evidence for consideration of including this in the nursing curriculum.
Response 5: Thank you for your valuable comments. We believe that Caring Massage (CM) needs more support and promotion so that it is considered at the international level.
Comments 6: Are the references appropriate?
References are thorough.
Response 6: Thank you for your valuable comments.
Comments 7: Any additional comments on the tables and figures.
Line 201 3,5- is this 3.5 or 3-5 years? I see this commonly throughout the manuscript. I'm just used to 3.5 vs 3,5. Page 10 is Bornout the same as Burnout or is this a typo?
Response 7: Thank you for your comments. We have made the corrections as suggested.
Comments 8: “Accept after minor revisions”.
The question was is this a language difference/preference. No other revisions were needed.
Response 8: Thank you for your time, every suggestion is valuable. We hope that the revisions have
improved the manuscript.

Reviewer 2 Report
Comments and Suggestions for Authors
Dear author,
There are many studies on massage, I suggest you explain the
contribution of this study to the literature in a better way. You can
increase the readability of the study by visualizing the tables
Author Response
REVIEWER_ 2
Comments 1: General Impression
There are many studies on massage, I suggest you explain the contribution of this study to the literature in a better way. You can increase the readability of the study by visualizing the tables.
Response 1: thank you for your time, every suggestion is valuable. It’s true, that there are many studies on massage but there is significant heterogeneity in the literature on various touch-massage techniques, benefits, and body areas where CM was administered. We want to shed more light on these aspects to support the importance of CM so that it is given more space in clinical care practice. We hope that the inserted definitions and the revisions have improved the manuscript.

Reviewer 3 Report
Comments and Suggestions for Authors
My most sincere congratulations to the authors of this study. The study itself is very meaningful to the nursing field as well as the field of integrative therapies, and it brings light into a very important topic which is the use of touch/massage interventions. Touch/massage interventions are within the most widely used integrative therapies in today’s world, and the most popular intervention about patients and professionals. The study is very well written and it was a pleasure to review.
The following are very minor suggestions, some typos and a few suggestions for better English translation. Looking forward to seeing such a meaningful publication online soon. Please see file attached with suggestions.

Author Response
REVIEWER_ 3
Comments 1: General Impression
My most sincere congratulations to the authors of this study. The study itself is very meaningful to the nursing field as well as the field of integrative therapies, and it brings light into a very important topic which is the use of touch/massage interventions. Touch/massage interventions are among the most widely used integrative therapies in today’s world, and the most popular intervention for patients and professionals. The study is very well written and it was a pleasure to review.
The following are very minor suggestions, some typos, and a few suggestions for better English translation. Looking forward to seeing such a meaningful publication online soon. Please see the file attached with suggestions.
Response 1: Thank you for taking the time to review the article, we appreciate your suggestions. The suggested changes certainly made a great contribution to the manuscript.

Reviewer 4 Report
Comments and Suggestions for Authors
This study was analyzed using the Delphi method. However, it was not conducted using Delphi analysis. Only the frequency values ​​for the items were presented.
In order to successfully conduct Delphi analysis, statistical figures such as content validity, convergence, consensus, and stability must be presented, and discussions must be presented along with the results.
First, content validity is important in the Delphi analysis technique. The content validity ratio is measured to determine whether the expert opinions have converged positively, but content validity was not presented in this study. In most studies, the formula for CVR (Content Validity Ratio) is used when using the Delphi analysis technique. Please present the CVR value of this study.
Second is convergence. Convergence is the agreement of experts’ opinions. Convergence must be less than .50. Please present the convergence.
Third is consensus. The agreement is an indicator of how much 50% of the experts responded in a certain section. The agreement should be greater than .75 to indicate that the experts' opinions are well agreed upon. Please provide the consensus.
Lastly, the stability coefficient. If the coefficient of variation is calculated by dividing the mean and the standard deviation, and if it is less than .50, the expert opinions are considered stable.
This stability coefficient must be calculated to determine whether the next Delphi survey will proceed based on this, or whether the expert opinions will be stably converged and the Delphi survey will be completed. Please provide the results for this.
The references were not written according to the submission guidelines of this society. Please correct them. In addition, more than half of the 30 references are old references that are more than 5 years old. Please correct the references while checking the contents of the text.
Author Response
REVIEWER_ 4
Comments 1: Delphi analysis.
This study was analyzed using the Delphi method. However, it was not conducted using Delphi analysis. Only the frequency values ​​for the items were presented. To successfully conduct Delphi analysis, statistical figures such as content validity, convergence, consensus, and stability must be presented, and discussions must be presented along with the results.
- First, content validity is important in the Delphi analysis technique. The content validity ratio is measured to determine whether the expert opinions have converged positively, but content validity was not presented in this study. In most studies, the formula for CVR (Content Validity Ratio) is used when using the Delphi analysis technique. Please present the CVR value of this study.
- Second is convergence. Convergence is the agreement of experts’ opinions. Convergence must be less than .50. Please present the convergence.
- Third is consensus. The agreement is an indicator of how much 50% of the experts responded in a certain section. The agreement should be greater than .75 to indicate that the experts' opinions are well agreed upon. Please provide the consensus.
- Lastly, the stability coefficient. If the coefficient of variation is calculated by dividing the mean and the standard deviation, and if it is less than .50, the expert opinions are considered stable. This stability coefficient must be calculated to determine whether the next Delphi survey will proceed based on this, or whether the expert opinions will be stably converged and the Delphi survey will be completed. Please provide the results for this.
Response 1: Thank you for your valuable comments. We conducted the Delphi analysis considering “Research Guidelines for the Delphi Survey Technique” of Hasson F., Keeney S. & McKenna H. (2000) according to which frequencies are converted into percentages and outcomes are considered relevant when the consensus reached ≥75%.
We also carried out the CVR evaluation for each item analyzed using the formula reported in various studies: CVR = (Ne - N/2) / (N/2), and we considered approximately 0.49 as the minimum CVR value for the large number of experts who participated in the study.
In each table, we have reported for each item the Content Validity Ratio (CVR), the mean (M), and the standard deviation (DS) with the relative coefficient of variation (CV) from which it is possible to see that the convergence and CVs are less than 0.50 and therefore, the opinions of the experts can be considered stable.
Regarding consensus, we reported for each item the % of experts who voted the item as "influential" and always corresponds to more than 50% of voting experts. But when this did not reach 75%, the item was eliminated.
We hope that the revisions have improved the manuscript.
Comments 2: References.
The references were not written according to the submission guidelines of this society. Please correct them. In addition, more than half of the 30 references are old references that are more than 5 years old. Please correct the references while checking the contents of the text.
Response 2: Thank you for your suggestions for improvement. We have re-evaluated the bibliography and made corrections. Our bibliography includes articles of different types on contact massage between nurse and patient. Given the particularity of the topic, we carried out a bibliographic search with a wide time to evaluate as many articles as possible.

Round 2
Reviewer 2 Report
Comments and Suggestions for Authors
A study that will contribute to the field
Author Response
Comments 1: General Impression
A study that will contribute to the field
Response 1: thank you for your time, every suggestion made has improved our manuscript.
Reviewer 4 Report
Comments and Suggestions for Authors
Thank you for your supplement to the previous review.
As I mentioned last time, Delphi analysis should present statistical figures such as content validity, convergence, consensus, and stability coefficient. Discussion should be presented along with the results. In the first revision, only content validity was presented. Content validity alone is insufficient as a content agreement value.
In addition, after content validity is analyzed, the results do not end with the first analysis, but require repeated revision. There is no part that was revised after reflecting and analyzing the results analyzed in the first round. Then, why should this study be conducted using Delphi?
In addition, Delphi analysis is analyzed by a group of experts to derive meaningful content, so in what part is the comparative analysis of patients and nurses necessary? It should be described what is academically valuable. There are many studies related to massage that have been published previously. Nevertheless, the readers should be able to understand why the comparative analysis was conducted on patients and nurses, not on a group of experts, and why the Delphi research method was used.
Lastly, a discussion and conclusion should be drawn on the derived CVR value.
This study conducted using the Delphi research method is insufficient. This content should be supplemented. It is judged that simply describing the contents of a questionnaire is not the Delphi method. Please first refer to other papers published using the Delphi research method.
Author Response
REVIEWER_ 4
Comments 1: Thank you for your supplement to the previous review.
As I mentioned last time, Delphi analysis should present statistical figures such as content validity, convergence, consensus, and stability coefficient. Discussion should be presented along with the results. In the first revision, only content validity was presented. Content validity alone is insufficient as a content agreement value.
Response 1: Thank you for your suggestions. We have changed and revised the data analysis according to your suggestion. We have inserted the consensus percentage, the CVR, the convergence, the IQR and mean and the SD. We would also underline that in previous research published on healthcare concerning the Delphi study, all these values have not been inserted (see https://www.mdpi.com/2227-9032/12/3/378, https://www.mdpi.com/2227-9032/12/22/2228, https://www.mdpi.com/2227-9032/12/20/2049, https://www.mdpi.com/2227-9032/12/19/1990, https://www.mdpi.com/2227-9032/12/13/1282, https://www.mdpi.com/2227-9032/12/7/739#app1-healthcare-12-00739...). We have also searched for figures in similar studies published by Healthcare or other journals (see https://journals.sagepub.com/doi/full/10.1177/00469580241246474, https://www.mdpi.com/2039-4403/14/3/185, https://onlinelibrary.wiley.com/doi/full/10.1111/jan.15064, https://www.sciencedirect.com/science/article/abs/pii/S1755599X20300392, https://pmc.ncbi.nlm.nih.gov/articles/PMC8371887/) and we do not find them. Tables are the best solutions to summarize Delphi results according to methodology manuscripts cited in our methods and also as suggested by Nasa, P., Jain, R., & Juneja, D. (2021). Delphi methodology in healthcare research: how to decide its appropriateness. World journal of methodology, 11(4), 116 and Holey, E. A., Feeley, J. L., Dixon, J., & Whittaker, V. J. (2007). An exploration of the use of simple statistics to measure consensus and stability in Delphi studies. BMC medical research methodology, 7, 1-10.
Comments 2: In addition, after content validity is analyzed, the results do not end with the first analysis, but require repeated revision. There is no part that was revised after reflecting and analyzing the results analyzed in the first round. Then, why should this study be conducted using Delphi?
Response 2: Thank you for your suggestion. We performed two rounds of Delphi. We have reported the results of these rounds and the reflection and analysis are divided according to the subsections: body area, patients’ outcomes and nurses’ outcomes.
Comments 3: In addition, Delphi analysis is analyzed by a group of experts to derive meaningful content, so in what part is the comparative analysis of patients and nurses necessary? It should be described as what is academically valuable. There are many studies related to massage that have been published previously. Nevertheless, the readers should be able to understand why the comparative analysis was conducted on patients and nurses, not on a group of experts, and why the Delphi research method was used.
Response 3: Thank you for your suggestion. The group of experts were only experts in the field of Caring Massage ®. This massage is a particular nursing approach, and we have justified in the introduction of the need for this Delphi study. We have not performed a comparative analysis, but we have studied which are the repercussions of the Caring Massage ® on the patients receiving it and, on the nurses, performing it. There is no comparison because there are different perspectives, from the users and the performers.
Comments 4: Lastly, a discussion and conclusion should be drawn on the derived CVR value.
Response 4: We have discussed the results following the CVR and CV in the first revision. The CVR and CV confirmed the precedent analysis that we simply made with percentage and mean and SD.
Comments 5: This study conducted using the Delphi research method is insufficient. This content should be supplemented. It is judged that simply describing the contents of a questionnaire is not the Delphi method. Please first refer to other papers published using the Delphi research method.
Response 5: Thank you for your suggestions to improve the manuscript. We have modified it accordingly and we hope that, similarly to previous research published on Healthcare, our manuscript is now suitable for publication.